# High-Performance Ni-SiC Coatings Fabricated by Flash Heating

Peter Renner, Ajinkya Raut ⬤ and Hong Liang *⬤

J. Mike Walker '66 Department of Mechanical Engineering, Texas A&M University,
College Station, TX 77843-3123, USA; parenner@tamu.edu (P.R.); ajinkyaraut@tamu.edu (A.R.)
* Correspondence: hliang@tamu.edu

**Abstract:** In this research, a novel flash heating coating application technique was utilized to create Ni-SiC coatings on carbon steel substrates with SiC contents much higher than is achievable using certain conventional coating techniques. Hardness profiles showed that the coatings improved the substrate by as much as 121%, without affecting the substrate. Tribotests showed that the wear performance was improved by as much as $4.7\times$ in terms of the wear rate ($mm^3/N \cdot m$) for the same coating when using an $Al_2O_3$ counterpart. Pure SiC coatings as a reference were also fabricated. However, the SiC coatings experienced elemental diffusion of Fe from the carbon steel substrate into the coating during fabrication. This occurred due to the increased heat input required for pure SiC to fuse to the substrate compared to the Ni-SiC coatings and resulted in decreased tribological performance. Diffusion of Fe into the coating weakened the coating's hardness and reduced the resistance to wear. It was concluded that ceramic–metallic composite coatings can successfully be fabricated utilizing this novel flash heating technique to improve the wear resistance of ceramic counterparts.

**Keywords:** Ni-SiC coatings; wear resistance; tribology; high hardness

## 1. Introduction

In various industrial applications, the typically used materials are not suitable for the environment they are placed in. For these situations, coatings are needed to supplement the surface properties of the base materials used. There are several advantages of using coatings, including reducing wear, friction, and corrosion by protecting the underlying substrate [1–5]. Coatings can also increase the hardness and fatigue resistance by preventing deformation [6–8]. By applying a coating, the favorable surface properties of the coating replace the underwhelming properties of the substrate. Coatings can be applied through several processes, such as electrocodeposition [9], electrophoretic deposition [10], and thermal spraying [11]. For these reasons, coatings are recommended for a wide range of applications [12–18].

Ni-SiC coatings represent some of the promising coatings for use in various demanding applications. The characteristics of wear resistance [19], hardness [20,21], and corrosion resistance [22,23] have been researched for Ni-SiC coatings. Since SiC is a second-phase material in such coatings, with the Ni acting as the matrix, the SiC particles tend to cause lattice distortions and reduce the crystal growth of the Ni, improving a variety of tribological characteristics [24–29]. Ni-SiC coatings have been shown to have higher hardness than pure Ni [25,30], can improve corrosion resistance when used coatings for certain materials such as magnesium alloys [3], and have better wear resistance than pure Ni [31]. Ni-SiC coatings are used in mechanical, chemical, petroleum, and protection applications [32], especially in the automotive [33] and manufacturing [34] industries. In both industries, equipment can vibrate, which can cause fretting, mean a coating such as Ni-SiC is often necessary. Some industrial applications also require operation in saline environments, which can cause corrosion damage. The combination of these factors can create an environment apt for the complex mechanisms of tribocorrosion [35], which Ni-SiC coatings are particularly well suited to handle [33].

There are several techniques used to produce ceramic–metallic composite coatings, such as Ni-SiC coatings. One of the more common methods is electrochemical deposition, often shortened to electrodeposition. In this method, an electrolyte is prepared with reagents added to distilled [36] or deionized [37] water. An electric field is applied to the electrolyte, causing the coating to deposit onto the substrate [38]. For Ni-SiC coatings, the SiC particles are immersed in the bath fluid [39]; however, there are some challenges in using this method. SiC particles dispersed in the bath fluid can cause uneven deposition in the coating, and the maximum SiC content depends on the SiC particle size [31]. Additionally, SiC particles are not easily embedded in coatings, meaning coatings typically have a low concentration of SiC particles. This issue can only be overcome using a surfactant or dispersant. These SiC particles can also be polished to prevent agglomeration and compaction [40]. Another disadvantage of this method is that the hardness of the coatings is not as high as the hardness produced by other methods [7,24,41]. A thermal technique for producing Ni-SiC coatings is thermal spraying, which involves melting the coating material at high temperatures and then spraying the melted material onto the substrate [11]. However, it is more expensive than other methods and requires much higher temperatures [42]. The thermal coating is affected by the morphologies of the Ni and SiC particles, although research has been performed to develop optimal particles for thermal spraying [43]. Thermal spraying is also a primarily mechanical method for applying coatings when it comes to metallic substrates, although the combustion gases may react with the substrate [44]. This method of application also suffers from the material cooling in the air before it strikes the substrate surface [45], although this can be mitigated using an environment other than air. This is due to thermal spraying systems often requiring a distance of several hundred millimeters between the spray nozzle and substrate surface, which in turn creates porous coatings [46–49]. The input process parameters of thermal spraying are also hard to utilize and result in low-quality coatings with large amounts of defects [44]. Silicon carbide also does not have a melting point under normal atmospheric conditions, sublimating at around 2500 °C, making it ill-suited for thermal spraying without an additional material such as Ni to act as the matrix [50]. Laser surface alloying is a widely accepted technique for generating Ni-SiC coatings [51,52]. This method utilizes high-power lasers to selectively melt a coating material. Due to the ability for localization, the substrate is effectively an infinite heat sync, which results in coating quenching. This process has similarities to the flash heating technique proposed in this research, despite the difference in heat source.

Flash heating is a relatively new technique that utilizes a high heat input to create high-melting-point coatings in a localized environment on low-melting-point substrates. In contrast to the mechanical bonds created by thermal spraying [44], with this method the coating primarily adheres to the material through metallurgical bonds [53]. Flash heating is also useful for joining materials that have incompatible crystal structures [54], when using an interlayer material with a crystal structure that is compatible with both the substrate and the coating [55]. Flash heating fabricates coatings with the nozzle at a distance of approximately 3 mm from the substrate [56] and creates fewer pores by effectively removing the particle cooling seen in thermal spraying [45,50,57]. Flash heating also results in no wasted material, being more cost-efficient than other methods. As a result, flash heating was chosen in this research to create Ni-SiC coatings with high SiC contents, in contrast to the low SiC contents generally studied in the literature [58–62].

## 2. Materials and Methods

### 2.1. Fabrication

To fabricate coatings by flash heating, an in-house-modified Eastwood 200 Amp tungsten inert gas (TIG) welder setup was used. The coating application process is shown in Figure 1. This flash heating coating process works somewhat similarly to wire arc additive manufacturing (WAAM). During fabrication, a voltage of 56 V under DC power is applied between the tip of a tungsten electrode and the ground. On top of the ground rests a substrate with a powder coating applied to its top surface, which is approximately

3 mm from the tungsten electrode. In this research, we used a protective argon gas with a flowrate of 15 cfh and a current of 110 A, which reached the electrode tip surrounded by an $Al_2O_3$ nozzle. The voltage combined with the current flow causes the protective gas to ionize, creating an electron or ion beam, which reaches temperatures in excess of 17,000 K for argon gas [63–65]. As this strikes the powder coating on the substrate's surface, it creates a localized melt pool almost instantly. Thus, the beam moves at a constant speed of 1–10 mm/s depending on the user input (5 mm/s in this work), effectively additively manufacturing the coating onto the substrate with a path width of 1–2 mm and track length of 8 mm per pass. Based off these values, if all energy enters the coating surface and assuming a perfectly circular melt pool, there is 1.96–7.84 kW/mm² of energy input into the coating depending on melt pool size. However, there will be some heat loss due to convection to the surrounding air, so these estimates are higher than the actual values.

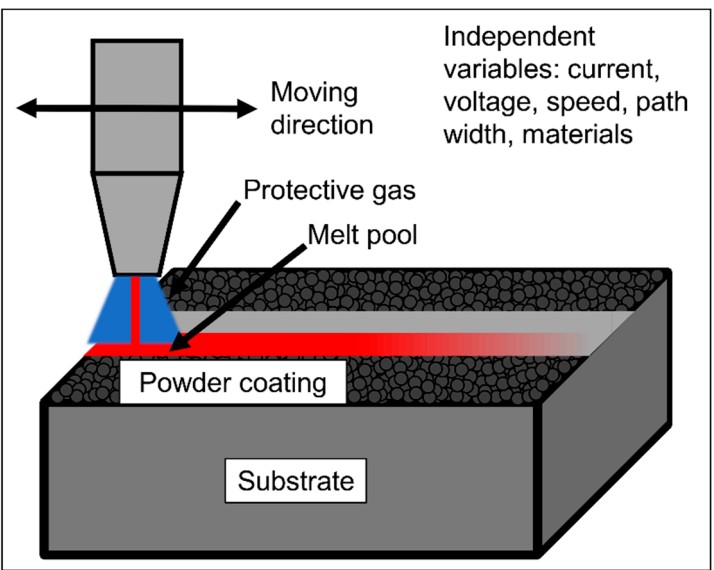

**Figure 1.** A schematic of the coating application process.

This flash heating additive manufacturing coating method has several advantages over other common coating techniques. The highly localized flash heating process minimally affects the substrate. Additionally, due to the substrate's large size relative to the localized melt pool, it effectively acts as a heat sink, similarly to the laser surface alloying process [51]. Thus, due to the large thermal gradient between the localized melt pool and substrate, the created coatings are quenched, which can improve a variety of tribological characteristics [66–68]. Flash heating of the coating utilizing this method also enables the fabrication of high-melting-point coatings, again due to the controlled localized regions that are heated. This coating application setup is also portable, so coatings can be applied on-site. Lastly, this method is cost-efficient due to no coating materials being wasted, with the only other consumable being the protective argon gas.

To fabricate these coatings, an ASTM A759 (quenched) carbon steel substrate was used with the composition shown in Table 1. This material is a high-hardness carbon steel designed for wear resistance and is commonly used in railroad rails. The substrates had a top surface of 10 mm × 10 mm and a thickness of 7.5 mm.

**Table 1.** Substrate composition in wt.%.

| C | Mn | P | S | Si | Cu | Ni | Cr | Fe |
|------|------|-------|-------|------|-----|------|------|---------|
| 0.81 | 0.98 | 0.012 | 0.013 | 0.28 | 0.3 | 0.11 | 0.23 | Balance |

Three coating compositions were studied in this research, as shown in Table 2. Ni (Sigma-Aldrich, Saint Louis, MO, USA), <50 μm, 99.7% pure, density = 8.91 g/mL) and SiC (Sigma-Aldrich, Saint Louis, MO, USA), 400 mesh, density = 3.22 g/mL) were used to fabricate these three coatings. High-SiC-content coatings were fabricated due to the advantages of this flash heating method, which were previously discussed. This is in contrast to the SiC content generally studied in the published research, reaching up to approximately 10 wt.% [58–62]. Prior to applying the powder coatings to the substrates, the powders were mixed at the respective ratios. The final thickness of each coating in this research was approximately 1.5 mm, although a wide range of thicknesses is possible with this coating application setup.

**Table 2.** The compositions of the three coatings analyzed in this research.

| Coating | Ni (wt%) | SiC (wt%) |
|---------|----------|-----------|
| 1 | 70 | 30 |
| 2 | 50 | 50 |
| 3 | 0 | 100 |

### 2.2. Characterization

Once coatings were fabricated, the top surfaces of all samples, including those of the substrates used for reference, were ground with SiC grit paper on a Struers DAP-3 polishing machine (Cleveland, OH, USA) at 300 RPM. Grits of 120, 240, 400, 800, and 1200 were used in successive order. A polishing pad with colloidal silica (Ted Pella, Inc., Middlefield, CT, USA), average particle size = 80 nm) was then used for chemical–mechanical polishing (CMP). CMP is a tribochemical polishing technique that utilizes the mechanisms of both mechanical wear and chemical degradation to refine surfaces to extremely low roughness [69–72]. For the cross-section analysis, samples were cut and then subsequently polished using the same grinding and polishing technique.

Once samples were polished, Knoop hardness tests were performed on each coating along with the substrate and reference material, namely annealed E52100 bearing steel. The Knoop hardness method creates an elongated diamond indentation and is similar to the Vickers method [73]. However, the Knoop method is advantageous, since its indentation results in a lower penetration depth than the Vickers method at equal loads. Additionally, Knoop indentations result in greater relief over Vickers indentations in terms of the resultant residual stresses [74]. The Knoop hardness *HK* can be calculated from Equation (1):

$$HK = 14,229 \times \frac{F_{(gf)}}{d^2_{k_{(\mu m)}}} \tag{1}$$

where $F_{(gf)}$ is the load applied in grams and $d^2_{k_{(\mu m)}}$ is the square of the length of the long diagonal created by the Knoop indenter measured in μm. Thus, a single measurement is required to calculate the Knoop hardness, whereas both diagonals must be measured to calculate the Vickers hardness [75]. This leads to lower relative error from Knoop hardness calculations than Vickers hardness calculations [76]. Additionally, due to the narrower indentations caused by Knoop hardness tests for the same load, indentations can be closer together, which is especially useful for cross-section characterization of coating–substrate interfaces [77].

Knoop hardness tests were performed on the top surfaces for all samples and on the cross-sections for all coatings. All tests were performed on four different samples to ensure statistical consistency. A Leco DM-400 LF Hardness Tester (Middlefield, CT, USA) with a diamond indenter was used for all tests, and a Leco Olympus PMG3 (Middlefield, CT, USA) inverted light microscope was used to subsequently image each indentation. For all Knoop hardness tests, a 300 gf load was applied over a period of 10 s for each indentation. For the top surface of each sample, 10 indentations were performed. For the

cross-section, tests were conducted every 0.25 mm, starting at 0.1 mm below the coating's top surface and continuing 3.5 mm down (15 total tests per sample) to fully characterize the coating–substrate hardness profile.

Tribotesting was also performed on the coatings along with the substrate. Wear tests were performed with a CSM Instruments tribometer in a linear reciprocating ball-on-plate orientation. These tests were performed under dry conditions with a 6.36 mm (0.25 in) $Al_2O_3$ ball acting as the wear counterpart. The $Al_2O_3$ balls had an average surface roughness Ra of 0.0762 μm. Each test used a load of 2 N, a wear track length of 6 mm, a distance of 120 m (10,000 laps), and a maximum speed of 5 cm/s. Acetone was used to clean each surface and remove any contaminants immediately prior to the start of each test.

After the completion of the wear tests, wear profiles were examined by interferometry. A Zygo NewView 600 Interferometer (Middlefield, CT, USA) was used with MetroPro 8.2.0 software (Middlefield, CT, USA). The lateral resolution was 1.10 μm and the vertical resolution was 0.1 nm. The vertical scan range was 150 μm. This setup was used to measure the average roughness of each polished sample prior to testing and to analyze the wear profiles after tribotesting. Table 3 shows the average roughness (Ra) and root mean square roughness (RMS) of the various coatings after undergoing CMP. Once the interferometry data were gathered for the wear profiles, the volumetric wear loss and wear rate were calculated using Matlab R2021a. A surface was formed from the xyz data of the interferometer using the Delaunay triangulation method [78,79]. This method creates a continuous surface made of triangles from the x, y, and z points. The volume enclosed between this surface and the z = 0 plane was calculated. The volume enclosed by the average z-height of the unworn surface and the z = 0 plane was subtracted from the volume between the continuous surface and the z = 0 plane to calculate volumetric wear loss. Scanning electron microscopy (SEM) imaging was also performed using a Tescan VEGA II SEM (Brno, Czech Republic) under the second electron detection mode using a voltage of 20 kV.

**Table 3.** The average roughness and RMS values of the three different coatings along with the substrate after CMP.

| Coating | Ra (μm) | RMS (μm) |
|---------|---------|----------|
| 1 (Ni-SiC 70-30) | 0.063 | 0.080 |
| 2 (Ni-SiC 50-50) | 0.060 | 0.083 |
| 3 (SiC) | 0.067 | 0.097 |
| Substrate | 0.056 | 0.073 |

## 3. Results

### 3.1. Increased Hardness Due to High-Hardness Carbides

The Knoop hardness tests were performed to compare the coatings fabricated in this research to both the substrate and the reference material of annealed E52100 bearing steel. Figure 2a shows indentations on the Ni-SiC 70-30 coating's surface with the typical elongated diamond shape, while Figure 2b shows the average and standard deviations of the hardness values for each material. Four samples of each material were tested with 10 tests per sample for a total of 40 tests per material. Figure 2b clearly shows that all coatings showed improved hardness over that of the substrate. The Ni-SiC 70-30 coating increased the average surface hardness by 61% relative to the substrate, the Ni-SiC 50-50 coating increased the same by 121%, while the SiC 100 coating only increased the average surface hardness by 40%. While flash heating can clearly improve the surface performance, for long periods of heat input, this begins to heat the substrate. Thus, for materials with a high melting point such as pure SiC, this process may cause elemental diffusion due to the substrate's temperature rising [80–83]. This is a topic for future work through heat transfer analysis. This elemental diffusion might be the cause of the trend seen for the hardness, whereby increasing the SiC content from 30 wt.% to 50 wt.% increases the hardness, al-

though the pure SiC coating performs closest to the carbon steel substrate, indicating that elemental diffusion reduces the performance of the coating. The addition of SiC in the Ni matrix has been shown to reduce the Ni crystal size by disrupting crystal growth, which is likely the primary cause for the greatly increased hardness values for the coatings [84]. Increasing the SiC content to 50 wt.% further increases the hardness, indicating that SiC further prevents Ni crystal growth and additionally plays a more primary role due to the high hardness of SiC.

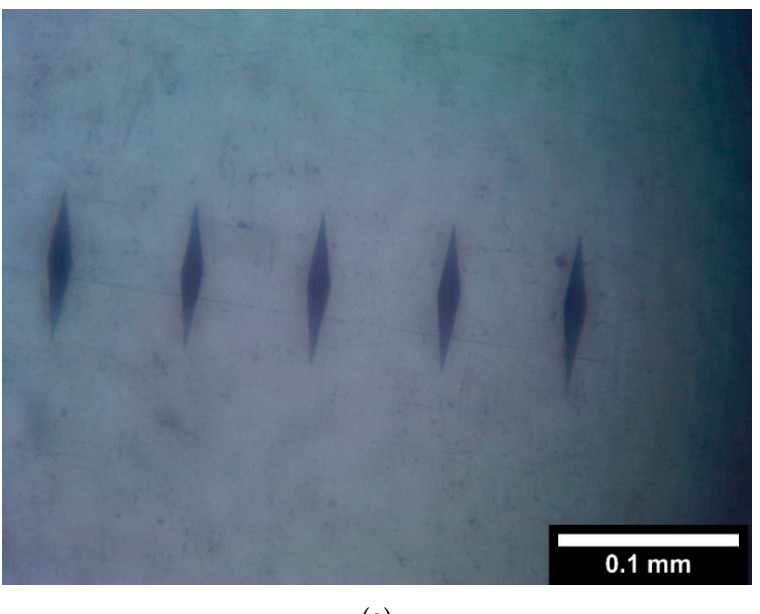

(**a**)

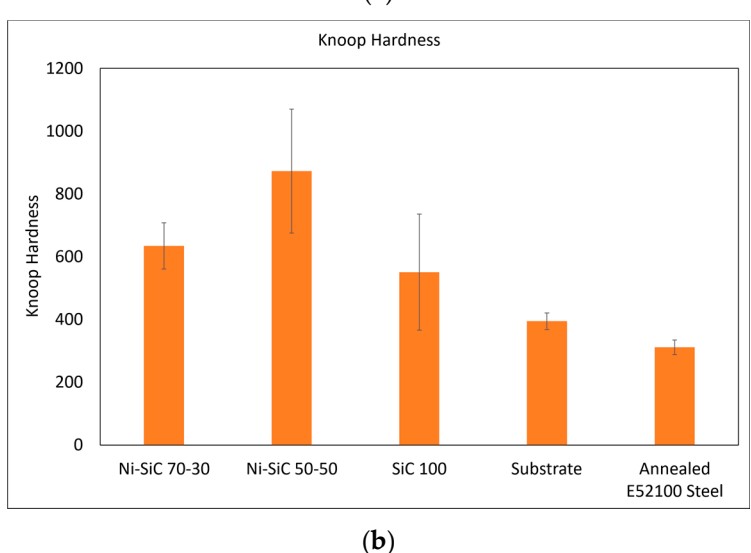

(**b**)

**Figure 2.** (**a**) Knoop hardness indentations on the Ni-SiC 70-30 surface taken under optical microscopy, (**b**) Knoop hardness values of the surfaces of the three coatings analyzed in this research, along with the substrate and the reference material of annealed E52100 bearing steel.

In addition to the surface hardness, the cross-section Knoop hardness of each coating was analyzed. To accomplish this, samples were first cut, mounted in epoxy, and then the same CMP process was followed as for the other samples. For each set of tests, each indentation occurred 0.25 mm downward from the coating's top surface than the previous test, and a total of 15 tests were performed, equating to 3.5 mm. Figure 3a–c shows the average and standard deviation of the cross-section hardness results for each coating. All three figures have the same *x*-axis and *y*-axis scales for comparison. In Figure 3a, the coating

(left side) has a Knoop hardness of approximately 600, while the substrate (right side) has a value of approximately 400, the same value as the surface hardness of the substrate seen in Figure 2b. Additionally, there is a clear difference between the improved coating hardness and substrate hardness. This shows that the Ni-SiC 70-30 coating greatly increased the hardness while minimally influencing the hardness of the substrate.

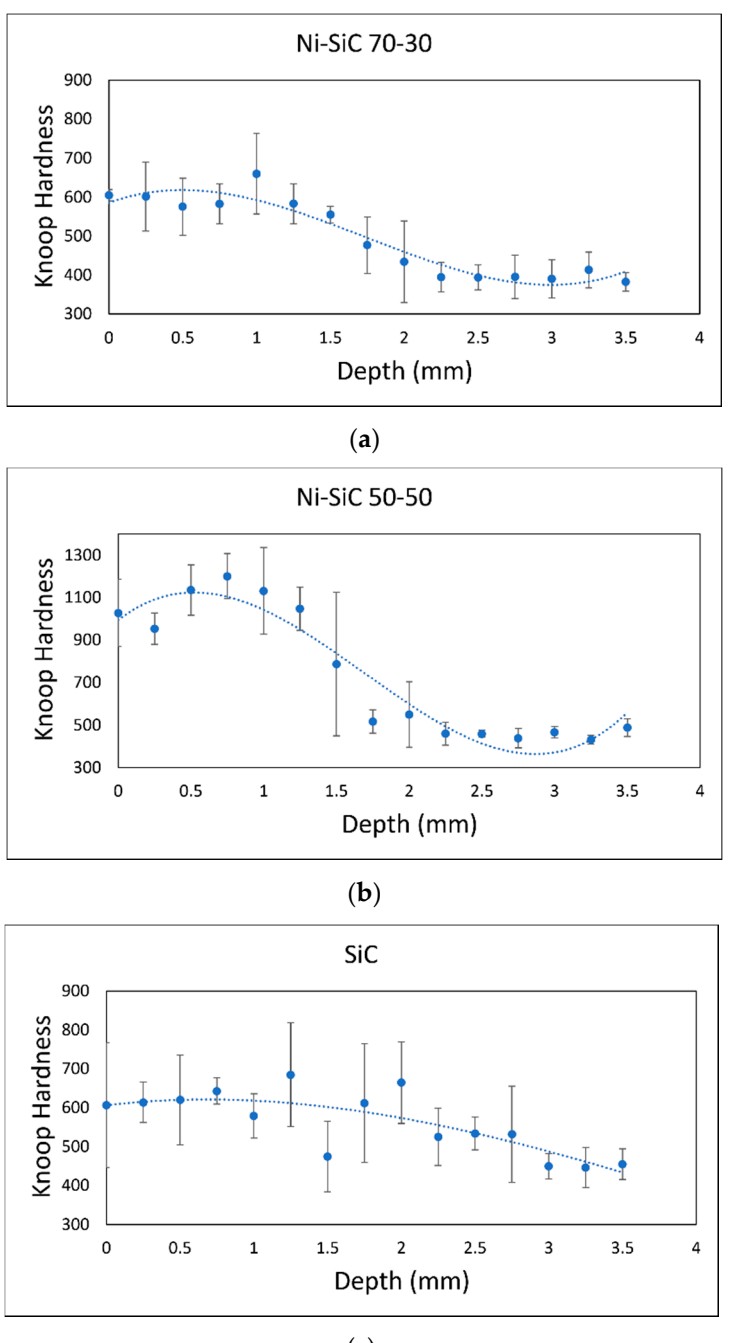

**Figure 3.** The average and standard deviation of the cross-section Knoop hardness tests for the (**a**) Ni-SiC 70-30 coating, (**b**) Ni-SiC 50-50 coating, and (**c**) SiC coating.

The difference between the coating and substrate is even more distinct for the Ni-SiC 50-50 coating, as seen in Figure 3b. Here, the coating's hardness is approximately 1100, while the substrate hardness is again approximately 400. Thus, increasing the SiC content from 30 wt.% to 50 wt.% further improves the hardness over that of the substrate. The standard deviations of the coating are larger than for the Ni-SiC 70-30 coating, which may

indicate variation in hardness between the samples. The Ni-SiC 50-50 coating may be easily influenced by variation in the heat treatment during flash heating, which could in turn affect the hardness of the coating. Nevertheless, the coating performs far better than the substrate, with hardness values comparable to high-hardness minerals such as topaz (8/10 on the Mohs hardness scale) and $\beta$-alumina [85]. The hardness at a depth of 1.5 mm has such a large standard deviation due to that point being before the coating–substrate interface (coating side) for some tests and after the interface (substrate side) for other tests, creating a large range of hardness values.

However, the pure SiC coating shows very different performance from the Ni-SiC 70-30 and Ni-SiC 50-50 coatings, as can be seen in Figure 3c. Here, there is no distinguishable difference between the coating and the substrate, especially when the standard deviations are taken into consideration. This may imply that the only cause of the different values is due to statistical variation between the various samples and the possibly heterogeneity of each sample. For instance, some hardness tests at the surface (furthest left point) showed a Knoop hardness of approximately 450, yet 2.75 mm beneath the surface the average hardness was 530, higher than that of the surface. This indicates again that elemental diffusion may have occurred due to the longer melting time of this coating, as discussed in Section 3.3. Many of the hardness values for Figure 3c are higher than the surface hardness of the substrate seen in Figure 2b. This may be due to the heat treatment increasing the hardness of the substrate during application of the SiC coating, or there may be diffusion of the carbon in the coating into the substrate, creating high-hardness carbides.

To verify the coating hardness values with depth, the hardness profiles shown in Figure 3 were overlayed onto SEM images of the coatings. Figure 4 shows this overlay for the three coatings. Generally, regions which appear brighter under SEM under second electron detection indicate lower electrical conductivity, while darker regions indicate increased conductivity. Since SiC is a semiconductor, it reduces the overall electrical conductivity of the coating relative to the carbon steel substrate. As such, in Figure 4 the coating is indicated by the brighter region while the substrate is indicated by the darker region. This is further verified through the rapid drop in hardness at 1.5–2 mm for the Ni-Sic 70-30 and Ni-SiC 50-50 coatings, where the coating–substrate interfaces are located. The hardness is relatively constant after this point, with values resembling that of the surface of the untreated substrate seen in Figure 2b. This indicates that the coating application has little to no effect on the substrate. For the SiC coating shown in Figure 4c, however, this change in brightness does not correspond to a change in hardness. This may be due to the increased porosity seen in Figure 4c in combination with Fe diffusion into the SiC coating. This porosity is only seen in the SiC coating, while the two Ni-SiC coatings exhibit minimal porosity.

### 3.2. Wear Performance

Wear tests were initially performed using cold-worked E52100 bearing steel balls as the counterparts during the tests, which is a common counterpart material used during tribotesting. Due to its high hardness, cold-worked E52100 bearing steel generally causes severe wear on test samples during tribotesting, although other parameters such as the load and speed also influence the results. However, as Figure 5 shows, the surface of the cold-worked E52100 bearing steel ball exhibited severe abrasive wear under the tribotesting parameters outlined in the experimental section. The circular region in this figure shows the flat surface formed on the ball as a result of wear, with the linear reciprocating motion in the horizontal direction. There is product buildup on the left and right sides of the circle, indicating that the ball experienced abrasive wear. As such, there was no distinguishable wear profile under interferometer for the coating surfaces when using a cold-worked E52100 bearing steel counterpart. To study the abrasive wear of the coatings, the $Al_2O_3$ ball mentioned in the experimental procedures was used instead. Figure 6 shows samples of the coefficient of friction properties for the substrate and three coatings for the duration of their tests.

Here, it can be seen that over the duration of the wear tests, the COF was relatively stable. There was a slight increase over the duration of the tests due to increased contact over time. The values were high in part due to $Al_2O_3$ generally having a high COF around 0.4–0.8 when under contact with common unlubricated materials such as alloys and ceramics [86–88]. After the wear tests were completed, the wear profiles were examined by interferometry. Figure 7 shows the wear profile for the substrate after wear.

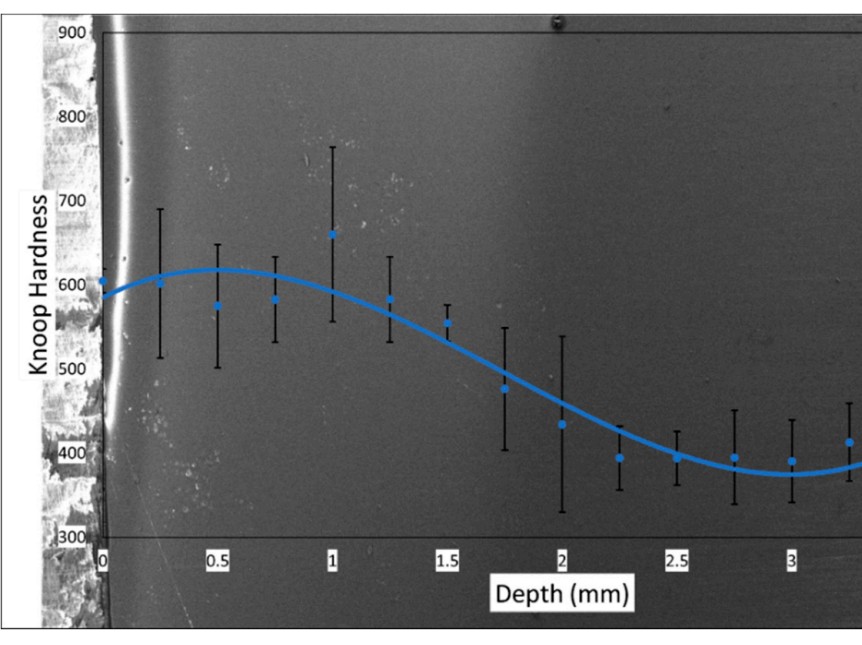

(**a**)

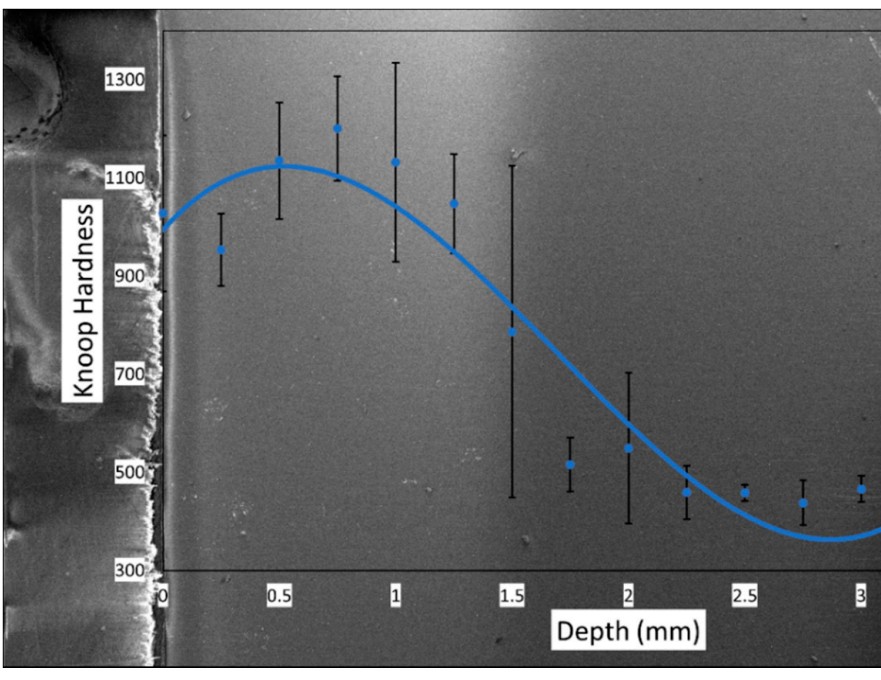

(**b**)

**Figure 4.** *Cont.*

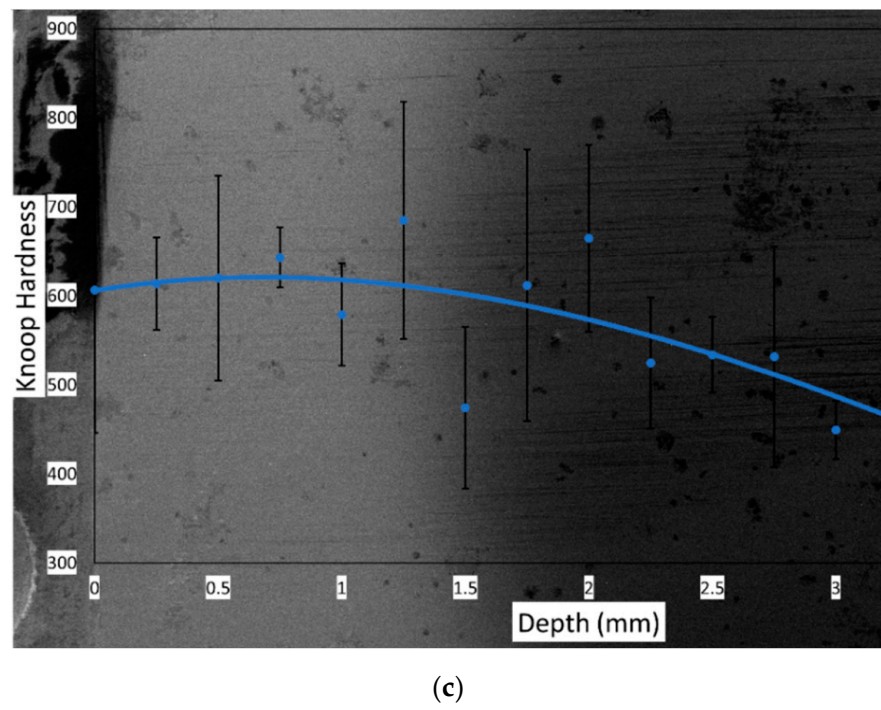

**(c)**

**Figure 4.** Cross-sections of (**a**) the Ni-SiC 70-30 coating, (**b**) Ni-SiC 50-50 coating, and (**c**) SiC coating with their hardness profiles overlayed to show hardness evolution into the substrate. The left side shows the coating surface in all images.

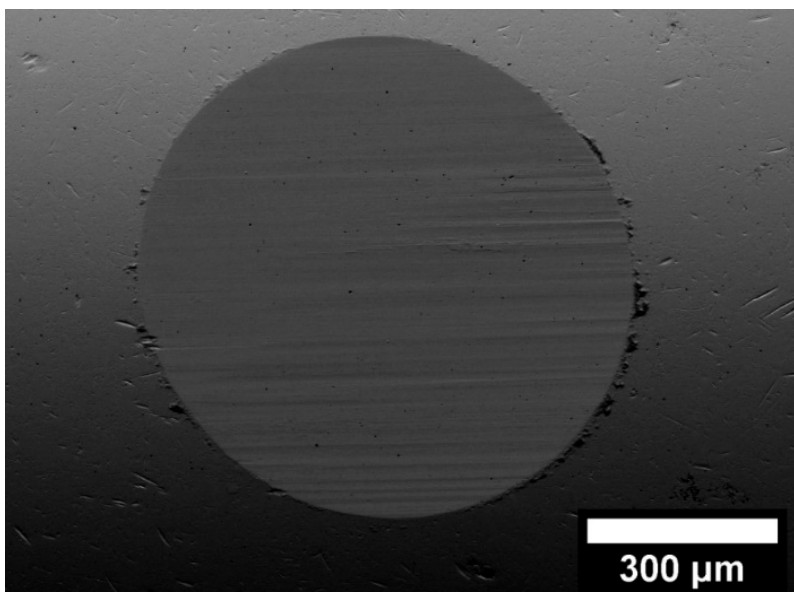

**Figure 5.** The surface of the cold-worked E52100 bearing steel ball initially used for tribotesting under SEM. The circle is a flat plane as a result of the wear the ball experienced.

In contrast to the substrate, all coatings had smaller wear profiles and increased wear performance over that of the substrate. These profiles can be seen in Figure 8. Here, the Ni-SiC 50-50 coating had the smallest wear profile and greatest performance, followed by the Ni-SiC 70-30 coating, with the SiC coating having a wear profile similar to that of the substrate.

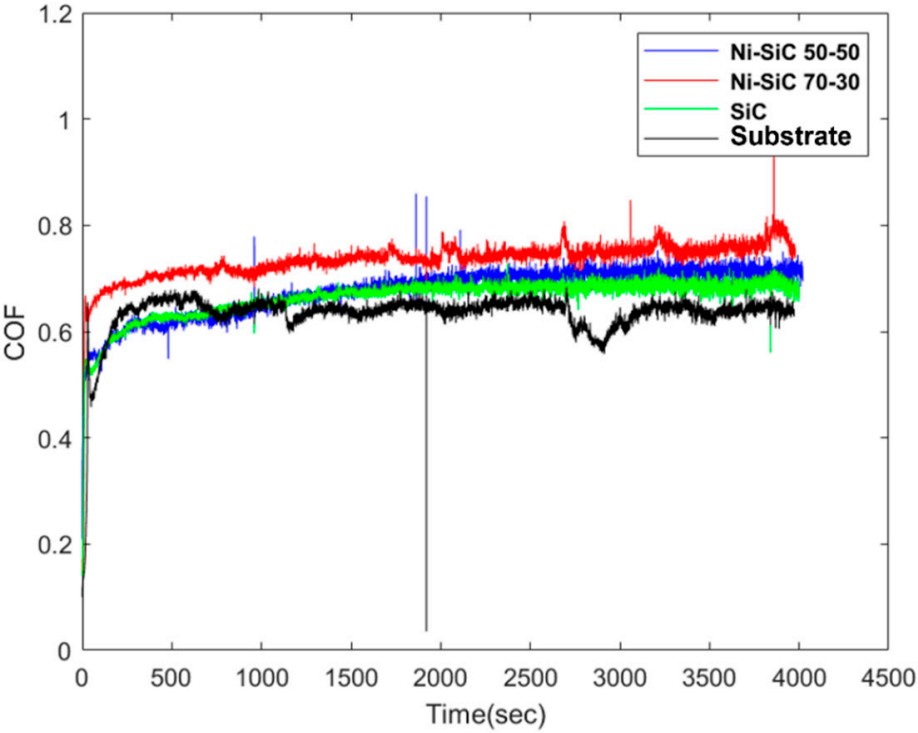

**Figure 6.** Coefficient of friction plotted against time during wear tests against an $Al_2O_3$ ball counterpart for the substrate and three coatings.

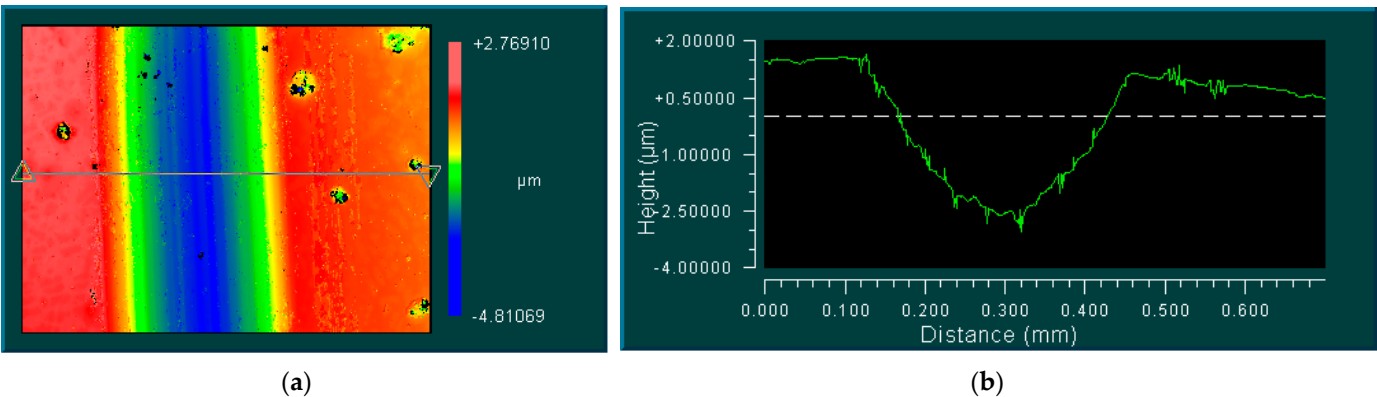

(**a**)                                                                           (**b**)

**Figure 7.** The substrate's (**a**) surface topography after the wear test and (**b**) cross-section profile at the location of the line shown in (**a**).

The wear profile data seen in Figures 7 and 8 were further analyzed to calculate the volumetric wear loss. The Archard and Hirst equation [89] was then used to calculate the wear rate, as shown in Equation (2):

$$K = \frac{Q}{Fs} \tag{2}$$

where $Q$ is the volumetric wear loss, $F$ is the normal load applied, and $s$ is the sliding distance. The calculated wear rates for the substrate and coatings are shown in Table 4. From Table 4, the trend for wear performance follows the trend for hardness seen in Figure 2b. These values show that the Ni-SiC 50-50 coating decreases wear by a factor of 4.71 in terms of the wear rate with respect to the substrate, while the Ni-SiC 70-30 coating wear reduction is 1.83 and the SiC coating wear reduction is only 1.18. This same trend was seen when the scratch performance of the same substrate coatings was analyzed [90].

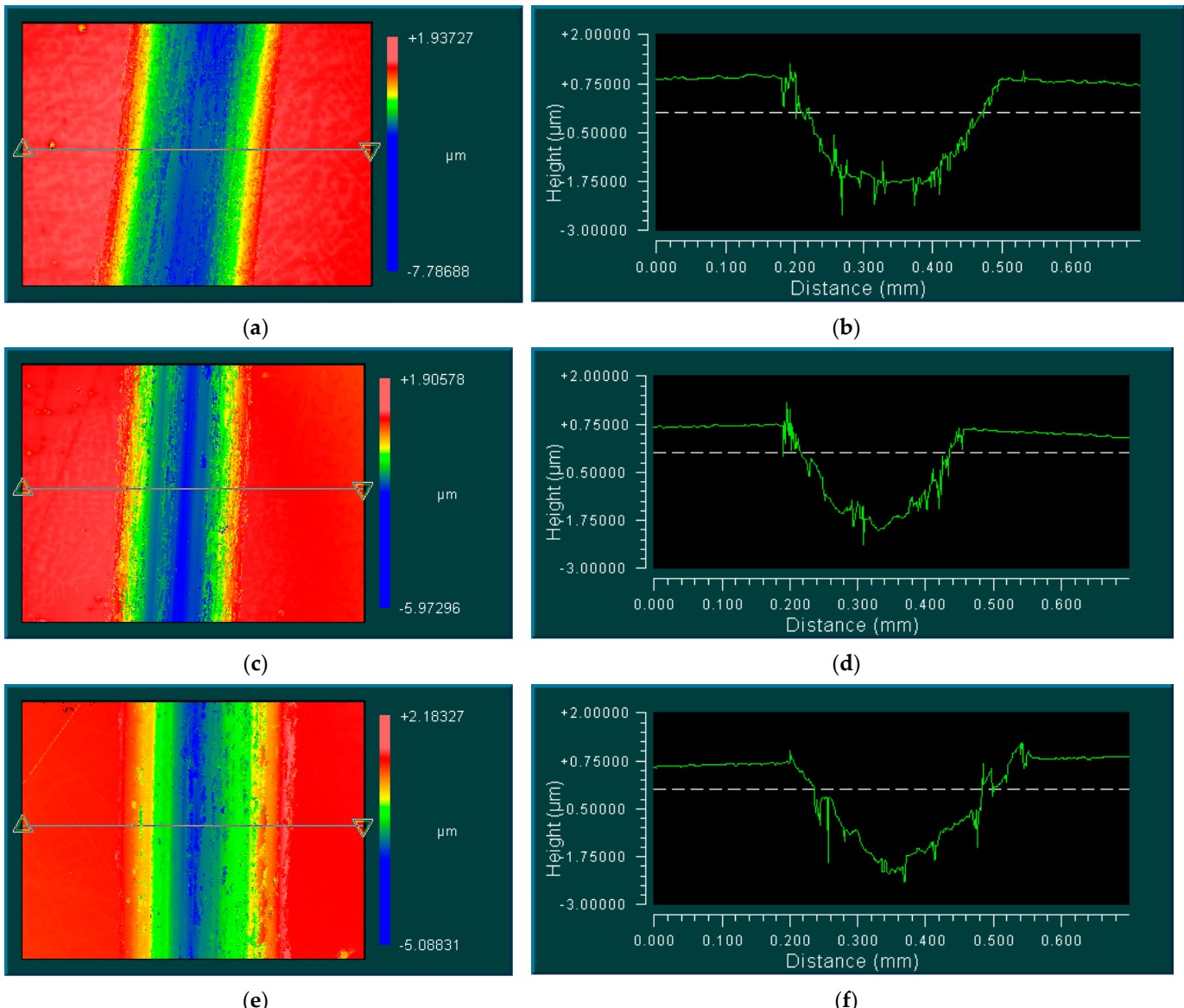

**Figure 8.** (**a**,**c**,**e**) The wear scar topography and (**b**,**d**,**f**) cross-section profiles of the Ni-SiC 70-30 coating, Ni-SiC 50-50 coating, and SiC coating, respectively.

**Table 4.** The wear rates for the substrate and coatings.

| Coating | Ni-SiC 70-30 | Ni-SiC 50-50 | SiC | Substrate |
|---|---|---|---|---|
| Wear rate (mm$^3$/N·m) | $2.91 \times 10^{-6}$ | $1.13 \times 10^{-6}$ | $4.53 \times 10^{-6}$ | $5.34 \times 10^{-6}$ |

Equation (3) below shows the Archard equation [91], an equation which Archard developed a few years prior to the Archard and Hirst equation shown previously:

$$Q = \frac{kFs}{H} \tag{3}$$

where *k* is a constant, which depends on several material properties and the environment, and H is the hardness of the weaker material. From Equation (3), the volumetric wear loss increases as the weaker material's hardness decreases. Figure 9 shows a plot of the log of the μm$^3$ wear volume versus hardness. The volumetric wear loss of the Ni-SiC 50-50 coating is lowest, followed by the Ni-SiC 70-30 coating and the SiC coating, while

the substrate has the highest volumetric wear loss. Meanwhile, the hardness follows the reverse trend, indicating that the Archard equation likely holds true. In terms of the wear rate, the Ni-SiC 50-50 coating and the Ni-SiC 70-30 coating also perform better than many ceramics when an $Al_2O_3$ counterpart is used in a pin-on-disk setup [92].

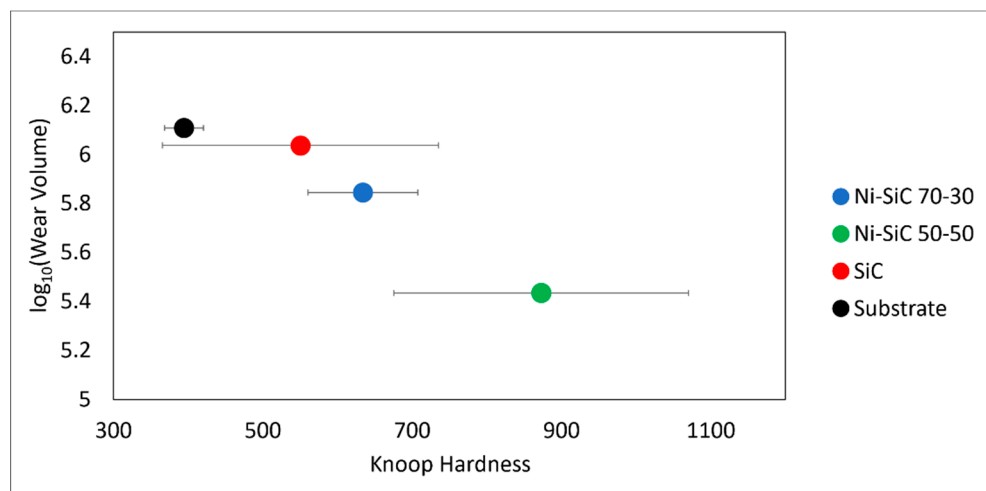

**Figure 9.** The base-10 log of the wear volume in $\mu m^3$ versus the Knoop hardness on the surfaces of the three coatings along with the substrate.

### 3.3. Mechanisms of Coating Formation

Due to the suboptimal performance of the SiC coating, EDS was used to analyze the elemental composition to determine whether elemental diffusion took place. This was accomplished using an FEI Quanta 600 FE-SEM with an Oxford energy dispersive X-ray spectroscopy (EDS) instrument. Figure 10 shows the EDS results for Si, C, Fe, and O, showing the four elements with the highest intensity in the SiC coating's top surface when analyzed via EDS. From the EDS analysis, the wt.% contents of Si, C, Fe, and O were 33.48, 35.83, 21.73, and 8.94, respectively, although a compositional analysis of EDS data is more of a qualitative than quantitative technique. The Si and C have the greatest intensity among the four elements, followed by Fe and then O. The Si and C are understandable due to those elements forming the composition of the coating, but the Fe can only be present as a result of elemental diffusion into the coating. This proves that elemental diffusion occurs and is the cause of the low tribological performance of the SiC coating. Additionally, the distributions of the Si and the O are similar, indicating that during flash heating the Si in the coating bonds with O in the air to form a type of silicon oxide. SiC can experience active or passive oxidation, depending on the environment, with temperature being the main factor [93,94]. At low temperatures when exposed to air, the reaction is as shown in Equation (4) [93]:

$$SiC(s) + \frac{3}{2}O_2(g) \rightarrow SiO_2(s) + CO(g) \tag{4}$$

The conversion from SiC to $SiO_2$ in the reaction above exhibits passive oxidation behavior and results in a mass increase at the surface [95]. However, when high temperatures are present, an active oxidation reaction takes place, as shown in Equation (5) [93]:

$$SiC(s) + O_2(g) \rightarrow SiO(g) + CO(g) \tag{5}$$

Since this second equation stipulates that the silicon oxide is in a gaseous form, this results in mass loss at the surface [96]. However, since oxygen is detected by EDS in Figure 10b and the distribution indicates it is bonded with Si, this oxidation reaction likely followed the first reaction. During flash heating, the sample was encompassed in inert

argon gas, likely preventing oxidation from occurring. Thus, this oxidation took place after the coating fabrication, after the sample was exposed to air for some time.

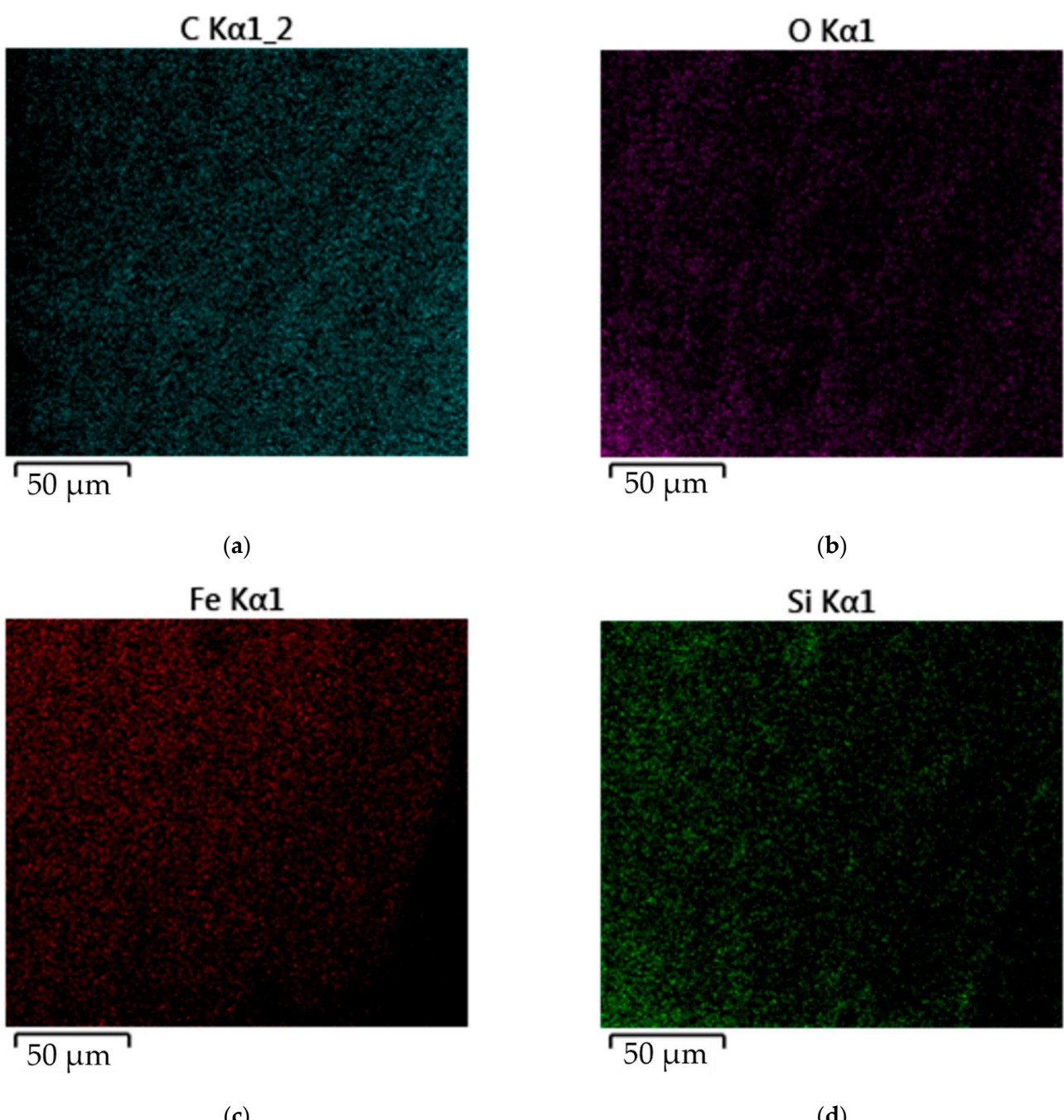

**Figure 10.** EDS results for the SiC coatings showing the elemental distributions of (**a**) C, (**b**) O, (**c**) Fe, and (**d**) Si. These were taken at the top surface of the SiC coating.

The Fe is clearly present in large quantities in the coating. Diffusion is a temperature-dependent process, as seen in Equation (6) [80–83]:

$$D = D_0 e^{-\frac{Q_d}{RT}} \qquad (6)$$

where $D$ is the diffusion coefficient, $D_0$ is a temperature-independent material constant, $Q_d$ is the activation energy for diffusion, $R$ is the gas constant, and $T$ is the temperature. An increased diffusion coefficient in a material increases the amount of diffusion that occurs.

Since T is in the denominator of the exponential, as temperature increases the diffusion coefficient D increases at an exponential rate. Thus, flash heating may cause issues by creating thick coatings (1.5 mm in this research) with materials that have high melting points without negatively influencing the substrate and causing diffusion. However, the addition of other materials (such as Ni in this research) can improve the melting time, resulting in the substrate not being heated by the flash heating process and meaning the coating's tribological characteristics are not negatively impacted. In certain scenarios, the substrate can also be water-cooled to prevent substrate heating.

To farther characterize the diffusion, Figure 11 shows an SEM image of the surface of the SiC coating. In this image, the light areas represent the SiC, while the dark areas represent the Fe. Some regions of Fe have bright areas, likely due to oxidation of the Fe to form nonconductive oxides at the surface. ImageJ was used to further analyze the image in terms of ratio of SiC to Fe. This was accomplished by applying a binary threshold using the Rényi Entropy method [97]. From this technique, the "analyze particles" option was used in ImageJ with circularity of 0–1. The results showed that the area percentages of SiC and Fe (and Fe oxides) were 58.22% and 41.78%, respectively. These values are relatively similar to the EDS compositional analysis results, indicating that this image likely does show the distribution of SiC and Fe.

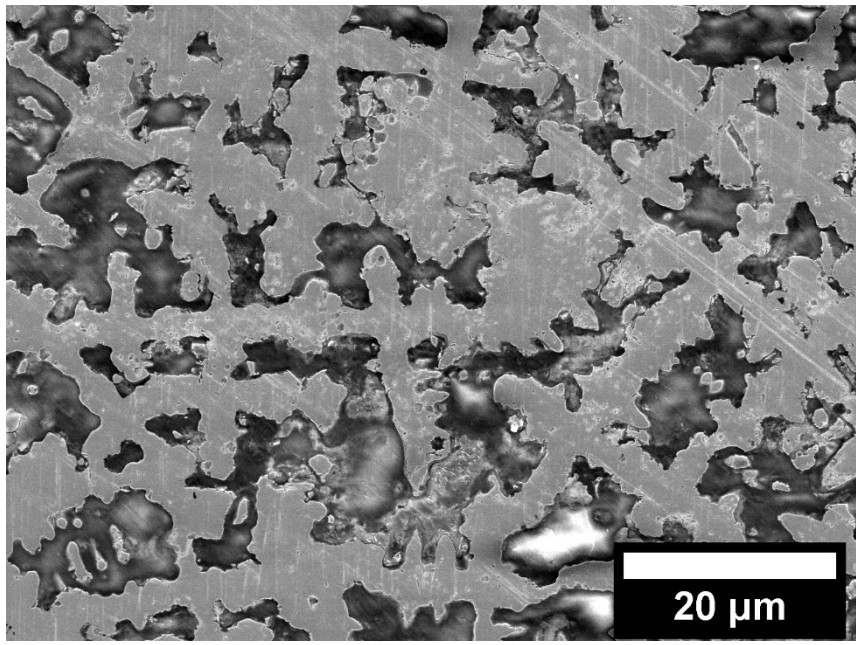

**Figure 11.** SEM image of the SiC coating's top surface.

Diffusion transports elements from high-concentration regions to low-concentration regions. As such, this analysis indicates that diffusion of Fe dominated during flash heating, resulting in a large quantity of Fe transferring from the substrate (high Fe concentration) to the coating (low Fe concentration). This process is shown in Figure 12a,b. Figure 12a shows an estimate of the distribution of Fe in terms of weight percent based on the known Fe content in the substrate of 97.265 wt.% and the measured Fe content at the surface from EDS of 21.73 wt.%. In Figure 12b, the red indicates increased temperature, while the arrows indicate the path of diffusion for Fe. Fe is the primary element that diffuses from the substrate due to it accounting for 97.265% of the substrate by weight. This diffusion of Fe may also cause some SiC particles to migrate into the substrate. Although the heat input is localized during flash heating, the energy transferred by heat is still large and rapidly increases the temperature of the substrate for the SiC coating. An increased diffusion coefficient such as is experienced by the substrate the SiC coating is applied to increase the number of Fe atoms diffusing into the SiC coating per unit time. While the time

during which this diffusion occurs is short due to the localized heat input, the temperature increase is great enough to cause substantial diffusion, which is unique to this flash heating procedure. As a result, the coating's composition is Fe-SiC instead of SiC.

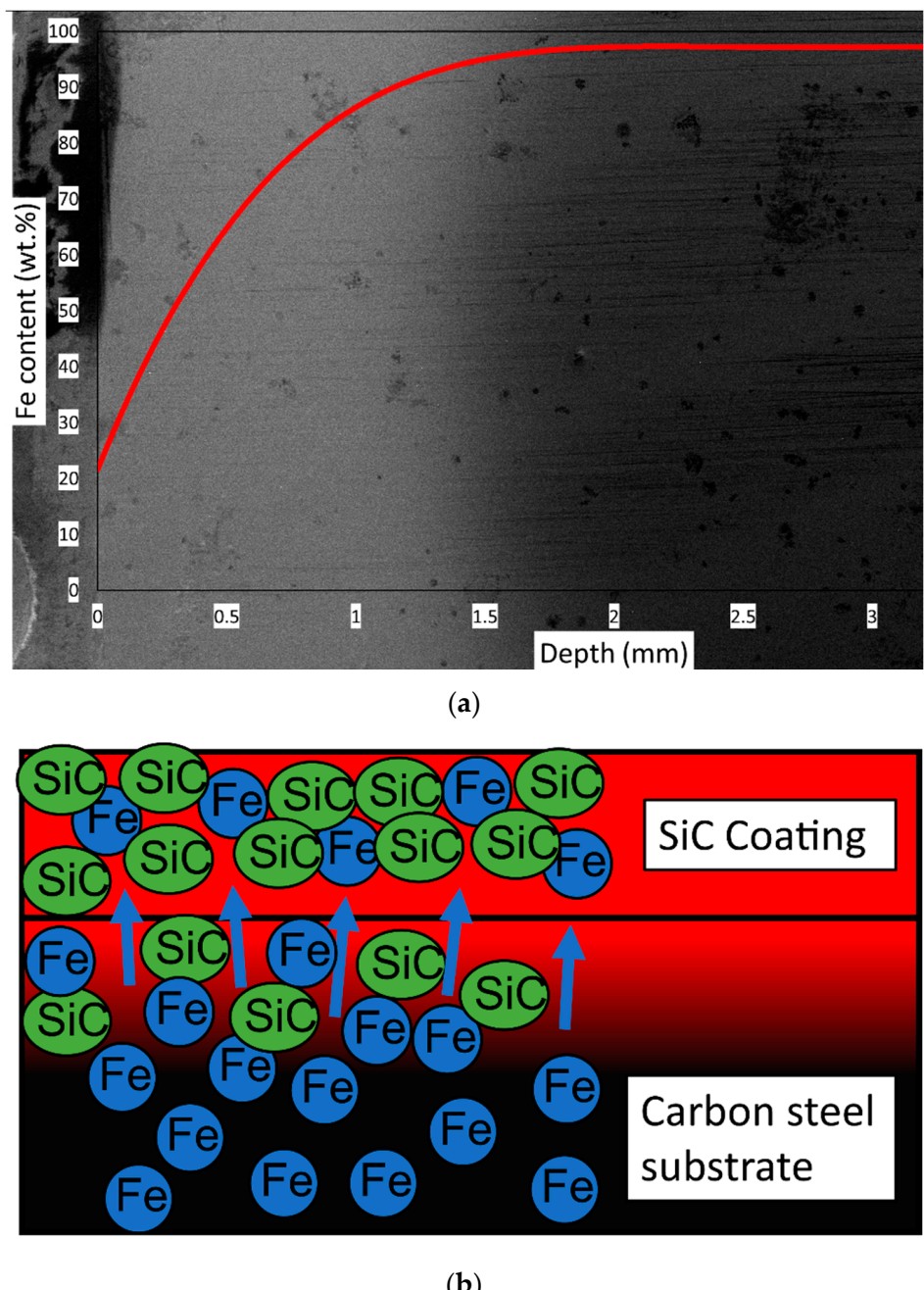

(**a**)

(**b**)

**Figure 12.** (**a**) An estimate of the Fe content distribution for the SiC coating and (**b**) the primary diffusion processes that occur in the SiC coating during Flash heating.

## 4. Conclusions

This research demonstrated the viability of flash heating as a means to fabricate Ni-SiC coatings with high SiC content. All coatings outperformed the carbon steel substrate during hardness tests. The Ni-SiC coating with 30 wt.% SiC improved the surface hardness by 61% compared to the carbon steel substrate, while the coating with 50 wt.% SiC improved the same parameter by 121%, with hardness values similar to superhard minerals such as topaz. Additionally, the hardness variation with depth showed that the coatings greatly improved

surface tribological characteristics while minimally affecting the substrate. Tribotests performed on the coatings and substrate also showed that the wear resistance levels of the coatings were greater than that of the substrate, with improvements of $1.83\times$ and $4.71\times$ for the Ni-SiC coatings with 30 and 50 wt.% SiC compared to the substrate, respectively. The pure SiC coating also improved both the hardness and wear over that of the substrate, but to a much more limited degree. This was due to elemental diffusion occurring during the flash heating process as a result of the increased required heat input for the SiC coating. Even though the time in which diffusion occurred was short, the temperature was high, which caused large amounts of Fe to diffuse due to the uniqueness of the flash heating process. This can be prevented by cooling the substrate during flash heating and with the fabrication of thinner coatings than were fabricated in this research.

**Author Contributions:** Conceptualization, P.R. and H.L.; experimentation: P.R. Topography: A.R. Data analysis, P.R and H.L. Manuscript writing and review: P.R. and H.L. project administration, H.L. All authors have read and agreed to the published version of the manuscript.

**Funding:** P.R. was funded by the National Science Foundation (NSF) Graduate Research Fellowship.

**Institutional Review Board Statement:** Not applicable.

**Informed Consent Statement:** Not applicable.

**Data Availability Statement:** The data presented in this study are available on request from the corresponding author. The data are not publicly available due to the associated project not being completed at this time.

**Conflicts of Interest:** The authors declare no conflict of interest.

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
