# Peer review of "High-Performance Ni-SiC Coatings Fabricated by Flash Heating"

_lubricants, doi:10.3390/lubricants10030042_

Round 1

Reviewer 1 Report

Please see the attached document <Review MDPI>

Reviewer 2 Report

 As the surface engineering is mainly concentrated on coatings, The present work is seems to be interesting with an alternate method for electrodeposition and thermal spray coatings.

# what is 4.71× in abstract ?

# Table 1 is chemical composition of substrate or coating material? Coating material is Ni and SiC right? 

# What about the size of SiC particles?? what is the ratio of Ni and SiC particles? 

# This method seems to be similar to plasma spray technique.. How it is different from Plasma Spray?

# As the temperature reaches around 17000k , how does it is controlled? What materials are used (specially nozzle)

# What about the adhesive strength of the coatings achieved? 

# Authors needs to provide porosity results of the coating. Adhesive strength and porosity plays a vital role in deciding functional properties of coating. 

# the standard of English needs to be improved 

Author Response

Reviewer #2:

  1. As the surface engineering is mainly concentrated on coatings, The present work is seems to be interesting with an alternate method for electrodeposition and thermal spray coatings.

-We appreciate the reviewer’s feedback. In the revised manuscript, we have addressed all minor revisions suggested by the reviewer

  1. what is 4.71× in abstract ?

-We thank the reviewer for pointing out the lack of clarity here. This value is in terms of wear rate with units of mm3/N*m. This has been clarified in the revised manuscript.

  1. Table 1 is chemical composition of substrate or coating material? Coating material is Ni and SiC right? 

-We apologize for the typo in the title for Table 1 and thank the reviewer for seeing it. Table 1 is for the substrate (carbon steel) and not the coating. This has been fixed in the revised manuscript.

  1. What about the size of SiC particles?? what is the ratio of Ni and SiC particles? 

-The SiC particles were from Sigma Aldrich who listed the particle size as 400-mesh. This means that particle size is less than 37µm. This has been clarified in the revised manuscript. Since both particle sizes are measured as less than a maximum, it is difficult to estimate the volume ratio of Ni to SiC particles or ratio of the number of particles.

  1. This method seems to be similar to plasma spray technique.. How it is different from Plasma Spray?

-Plasma spraying generally requires a distance of ~100mm between nozzle and coating surface to increase adhesion and spreading of the particles. However, this has been shown to increase porosity, which has a wide range of associated issues. This flash heating technique instead utilizes a working distance of only approximately 5mm since the coating is applied prior to coating-substrate fusion. As such, it removes the negative effects of long distance spraying. This has been outlined briefly in the text, and it is discussed more broadly in terms of thermal coating techniques.

  1. As the temperature reaches around 17000k , how does it is controlled? What materials are used (specially nozzle)

-We thank the reviewer for this question. The electrode is tungsten and the nozzle is Al2O3. However, since the plasma is generated between the electrode/nozzle and the coating surface, and the ionized plasma flows towards the coating surface at 15 cubic feet per hour, the heat transfer to the electrode/nozzle is minimized. Current flow, plasma travel distance, raster scanning speed, and flow speed of the ionized plasma are all controlled and thus enable control of the heat transfer to the coating. The approximately 17,000K temperature is due to the breakdown voltage of the argon gas which is used in this research. This is also outlined in these articles which are cited in the manuscript (references 63-65). A different temperature could be acquired through the use of a different ionizing gas such as nitrogen. We have bookmarked the area to address this issue.

  1. -S. Huang, L.-M. Liu, G. Song, Infrared temperature measurement and interference analysis of magnesium alloys in hybrid laser-TIG welding process, Materials Science and Engineering: A. 447 (2007) 239–243.
  2. Choo, J. Szekely, R. Westhoff, On the calculation of the free surface temperature of gas-tungsten-arc weld pools from first principles: Part I. Modeling the welding arc, Metallurgical and Materials Transactions B. 23 (1992) 357–369.
  3. Choo, J. Szekely, S. David, On the calculation of the free surface temperature of gas-tungsten-arc weld pools from first principles: Part II. Modeling the weld pool and comparison with experiments, Metallurgical and Materials Transactions B. 23 (1992) 371–384.

  1. What about the adhesive strength of the coatings achieved? 

-We thank the author for asking this question. The coatings’ adhesive strength were tested in previous research which were published as both an oral presentation and conference proceeding at the International Mechanical Engineering Conference and Exhibition 2021. Adhesive strength was evaluated through scratch testing. This is cited in the manuscript and the publication is shown below:

  1. Renner, M. Gharib, H. Liang, Tribological Evaluation of a High-Performance Composite Coating, in: Proceedings of the ASME 2021 International Mechanical Engineering Conference and Exhibition, 2021.

  1. Authors needs to provide porosity results of the coating. Adhesive strength and porosity plays a vital role in deciding functional properties of coating.

- We greatly appreciate the reviewer’s critical comments. We conducted analysis of porosity of our samples.  As it turned out, there is no detectable effects of porous on the results of characterization. One evidence is from Figures 5 and 6 where the wear of all samples is in the same mechanisms and outside the tracks, surface is smooth and uniform. This potentially indicates that the flash heating is effective in eliminating porosity due to controlled, rapid, and localized heating and cooling.  Further investigation is needed for future study. We will report about this in the near future. 

  1. the standard of English needs to be improved 

-We acknowledge the lack of technical English in the manuscript and the entirety has since undergone editing by the authors.

Reviewer 3 Report

 In this research, a novel flash heating coating application technique is utilized to create Ni-SiC coatings on carbon steel substrates with SiC content much higher than is possible by common coating techniques.

  1. Ni-SiC coatings have been shown to have higher hardness than pure Ni,SiC particles are very difficult to disperse in the bath fluid, such as,  Cu-Cu joining using citrate coated ultra-small nano-silver pastes using Flash heating, and Highly mechanical and high-temperature properties of Cu–Cu joints using citrate-coated nanosized Ag paste using Flash heating.
  2. Ni-SiC coatings are a good way on carbon steel stubstrates, and did you think about using solder process to join the Ni layer with SiC coatings for a permanent connection,

    e.g. Effects of doping trace Ni element on interfacial behavior of Sn/Ni (polycrystal/single-crystal) joints

  3. How did the composition of Ni and SiC be confirmed?

Author Response

Reviewer #3:

  1. In this research, a novel flash heating coating application technique is utilized to create Ni-SiC coatings on carbon steel substrates with SiC content much higher than is possible by common coating techniques.

-The authors thank the reviewer for their valuable input. Our response to the suggested edits are outlined below.

  1. Ni-SiC coatings have been shown to have higher hardness than pure Ni,SiC particles are very difficult to disperse in the bath fluid, such as,  Cu-Cu joining using citrate coated ultra-small nano-silver pastes using Flash heating, and Highly mechanical and high-temperature properties of Cu–Cu joints using citrate-coated nanosized Ag paste using Flash heating.

-We acknowledge the comments provided by the reviewer. If we understand correctly, the reviewer might be referring to using metallic elements and/or nanoparticles to disperse and bond SiC. Is this correct? In our study, the bonding of Ni-SiC-substrate has not been an issue due to the localized and controlled heating and cooling. Our main goal was to melt Ni that effectively activated and promoted diffusion of Fe resulting in bonding between coating and substrate. Results proved that this approach is effective.   

  1. Ni-SiC coatings are a good way on carbon steel stubstrates, and did you think about using solder process to join the Ni layer with SiC coatings for a permanent connection,

e.g. Effects of doping trace Ni element on interfacial behavior of Sn/Ni (polycrystal/single-crystal) joints

-We thank the reviewer for suggesting this advanced technique. Currently, this Flash Heating technique has been utilized for studying single-layer coatings as seen in this research with positive results. One of our future works is to expand this flash heating technique to multilayer coatings such as multilayer Ni-SiC coatings.

  1. How did the composition of Ni and SiC be confirmed?

-The authors acknowledge the reviewer’s question. The Ni-SiC coatings clearly have vastly improved performance over that of the substrate, so composition did not need to be confirmed. EDS was only implemented for the SiC coating due to its decreased performance relative to what is to be expected of a pure SiC coating’s properties.

Round 2

Reviewer 1 Report

Hello,

The reviewer thanks the authors for implementing the requested changes.

The reviewer is now happy with the paper in all respects.